# Distinct melanocyte subpopulations defined by stochastic expression of proliferation or maturation programs enable a rapid and sustainable pigmentation response

Ayush Aggarwal[1,2], Ayesha Nasreen[1,2], Babita Sharma[1,2], Sarthak Sahoo[3], Keerthic Aswin[1,2], Mohammed Faruq[1,2], Rajesh Pandey[1,2], Mohit K. Jolly[3], Abhyudai Singh[4,5], Rajesh S. Gokhale[6,7], Vivek T. Natarajan[1,2]*

1 CSIR-Institute of Genomics and Integrative Biology, New Delhi, India, 2 Academy of Scientific and Innovative Research (AcSIR), Ghaziabad, India, 3 Department of Bioengineering, Indian Institute of Science, Bangalore, India, 4 Electrical and Computer Engineering, University of Delaware, Newark, Delaware, United States of America, 5 Biomedical Engineering, University of Delaware, Newark, Delaware, United States of America, 6 National Institute of Immunology, New Delhi, India, 7 Indian Institute of Science Education and Research Pune, Pune, India

* tnvivek@igib.in

**Data Availability Statement:** All the sequencing data generated in this study is available at GEO under the accession GSE233198 superseries.

## Abstract

The ultraviolet (UV) radiation triggers a pigmentation response in human skin, wherein, melanocytes rapidly activate divergent maturation and proliferation programs. Using single-cell sequencing, we demonstrate that these 2 programs are segregated in distinct subpopulations in melanocytes of human and zebrafish skin. The coexistence of these 2 cell states in cultured melanocytes suggests possible cell autonomy. Luria–Delbrück fluctuation test reveals that the initial establishment of these states is stochastic. Tracking of pigmenting cells ascertains that the stochastically acquired state is faithfully propagated in the progeny. A systemic approach combining single-cell multi-omics (RNA+ATAC) coupled to enhancer mapping with H3K27 acetylation successfully identified state-specific transcriptional networks. This comprehensive analysis led to the construction of a gene regulatory network (GRN) that under the influence of noise, establishes a bistable system of pigmentation and proliferation at the population level. This GRN recapitulates melanocyte behaviour in response to external cues that reinforce either of the states. Our work highlights that inherent stochasticity within melanocytes establishes dedicated states, and the mature state is sustained by selective enhancers mark through histone acetylation. While the initial cue triggers a proliferation response, the continued signal activates and maintains the pigmenting subpopulation via epigenetic imprinting. Thereby our study provides the basis of coexistence of distinct populations which ensures effective pigmentation response while preserving the self-renewal capacity.

Publicly available data reanalysed as part of this study is submitted under the accessions GSE151091 and EGAS00001002927. All the processed files of the single cell data generated in this publication are available as RDS files on Zenodo (https://doi.org/10.5281/zenodo.12536179). The RDS files associated with single-cell RNA sequencing can be uploaded and easily explored on an interactive web application https://ayushagg.shinyapps.io/MelDat/ developed by us.

**Funding:** VTN secured funding for this project. The Grant number is MLP2008 (project Regen-X). Funder - Council of Scientific and Industrial Research (CSIR) URL of the Funder - https://www.csir.res.in/ The funders had no role in study design, data collection and analysis, decision to publish, or preparation of the manuscript.

**Competing interests:** RSG is the co-founder of Vyome Biosciences Pvt Ltd., a biopharmaceutical company working in the area of dermatology. Other authors declare no competing interest.

**Abbreviations:** cAMP, cyclic adenosine monophosphate; DAR, differentially accessible region; DBPS, Dulbecco's phosphate buffer saline; DHAcR, differentially hyperacetylated region; DMEM, Dulbecco's Modified Eagle's Media; FBS, fetal bovine serum; GRN, gene regulatory network; hpf, hours post fertilization; IF, immunofluorescence; IFN, interferon; NHEM, normal human epidermal melanocyte; ODE, ordinary differential equation; PMSF, phenyl methyl sulphonyl fluoride; TF, transcription factor; UV, ultraviolet.

## Introduction

Melanocytes are specialised skin cells responsible for producing pigment, which serves as a defence mechanism against harmful UV radiation [1–4]. When human skin is exposed to UV rays, melanocytes activate 2 protective responses: stress response which includes proliferation [5] and pigmentation, to prepare the cells for future exposure [6]. Intriguingly, both of these programs are triggered by the same signal and regulated by the dynamics of MITF [7]. In an earlier study, it was shown that dedifferentiation is associated with cell proliferation. Herein most pigmented clones were small and most unpigmented clones large [8]. However, it remains unclear how melanocytes maintain both these programs within the same population. Moreover, the classical notion is that the human epidermis consists of only the terminally differentiated state of melanocyte. However, recent studies have challenged this notion.

For instance, a study identified a stem cell population in the interfollicular epidermis that can differentiate into pigmented melanocytes [9]. Another study showed that melanocytes derived from developmentally distinct iPSC-derived progenitors differ in their pigmentation, migration, and proliferation potential [10]. In certain altered states of the skin, such as hypertrophic scars following burn injury, amelanotic melanocytes have been observed, which can potentially become pigmented upon treatment with alpha melanocyte stimulating hormone (α-MSH) [11]. Three separate studies in the recent years have shown the diversity and plasticity of the melanocyte population in the hair follicle niche [12–14]. These studies suggest the existence of melanocyte states other than the terminally differentiated pigmented state in the epidermis and especially in the hair follicle (which is shielded from UV). This hair follicle cycle serves as a trigger for the interconversion of these states under the defined program controlled by the niche. However, the coexistence of such transcriptionally distinct melanocyte subpopulations in the epidermal skin is not yet known, and if they do exist, whether they can interchange between states upon environmental triggers such as UV remains speculative.

We hypothesise that the diversity inherent in a seemingly uniform population can facilitate rapid adaptation by enabling swift interchangeability when needed. Advancements in single-cell sequencing (scRNA-seq) technologies have provided evidence of heterogeneity within previously thought homogeneous cell populations [15]. This heterogeneity possibly allows the cells to respond differentially to external cues according to their specific needs. While genetic mutations were initially attributed as the primary source of heterogeneity, nongenetic factors such as inherent gene expression variability are emerging as important determinants of cellular diversity [16–19]. Such variability in intrinsic cell states is proposed to be an inherent feature that enables faster and more efficient responses to external cues [18].

In this study, we utilise single-cell-based methodologies to identify transcriptional and phenotypic states of melanocytes. Our findings demonstrate that the human and zebrafish skin contains proliferation and pigmentation-competent melanocyte states, which can be captured under culture conditions. Using a temporally resolved progressive pigmentation model, we observe stochasticity in the attainment of high- and low-pigmented (proliferative) states. Through single-cell RNA sequencing (scRNA-seq), we observe a sequential transition of population dynamics over time and identify the preexistence of proliferation and pigmentation programs in separate subpopulations. Manipulating the α-MSH pathway using a pharmacological activator alters the dynamics of these states, shifting melanocytes towards either proliferative or pigmented states. A GRN derived from active enhancers in the 2 states independently simulates the coexistence of these 2 melanocyte states. Thereby, we establish a conserved network that supports the coexistence of 2 distinct melanocyte states, differing in proliferation and pigmentation, using 2 independent cell-based model systems.

## Results

### Pigmentation and proliferation programs are maintained in distinct subpopulation within the melanocyte pool

To understand how melanocytes maintain the divergent programs of pigmentation and proliferation within the population, we reanalysed the NHEM scRNA data from GSE151091 [20]. After the initial dimensionality reduction and clustering of the melanocyte subset using Seurat [21], we observed that age group, donor ID, or plate ID were a possible source of heterogeneity that was reflected in the UMAP space (S1A Fig). While Belote and colleagues [20] identified the age and site-specific differences within the human melanocyte pool, we were interested in looking beyond and identifying functionally distinct melanocyte states, if any, irrespective of their location in the body or the age group from which they were isolated. Hence, we regressed out these effects using Harmony [22] and identified 5 distinct clusters (S1B Fig), possibly representing distinct functional states of melanocytes. Cluster markers and their functional annotation revealed a mature cluster, which showed high expression of *MITF*, *TYR*, and *DCT*, a proliferative cluster which showed enrichment of *MKI67* and *TOP2A*, and an immature population having expression of both proliferation and pigmentation genes. A stem cell-like cluster enriched for genes such as *TWIST1*, *TWIST2*, and *AXL*. Interestingly, 1 cluster showed enrichment of interferon (IFN) signalling genes, such as *IFIT1*, *IFIT2*, and *STAT1* (Figs 1A, S1C, and S1D).

While studies have reported IFN gene signatures in melanocytes [23,24], especially in disease conditions such as vitiligo [25], this is the first time a stable state enriched for interferon signatures has been identified in melanocytes from human skin. While the majority of the mature and the IFN-enriched cells came from the adult sample, the stem cell-like and the proliferative cells were enriched in the foetal skin. Although the proportions of these subpopulations were different, these states were present across all age groups (S1E Fig). This analysis corroborates Belote and colleagues' [20] findings and suggests the existence of multiple melanocyte cell states in the human epidermal skin. The same could be observed in another publicly available human melanocyte data set from adult epidermal cells derived from human skin, EGAS00001002927 [26] (S2A–S2D Fig). Therefore, it is tempting to speculate that melanocytes, much like immune cells and neurons, can exhibit distinct functional states in the same environment [27,28].

These transcriptionally distinct clusters are deduced from cross-sectional single-cell data sets from the human epidermis. This raises the following questions: Do these subpopulations coexist in the same environment? Does each cell in these clusters transition between these states following a set trajectory? Or are these states stable and self-sustaining in nature?

To confirm the coexistence of these states in another in vivo environment we utilised the zebrafish model system. We isolated melanocytes from zebrafish embryos at 24 hours post fertilisation (hpf) and conducted scRNA-seq. Our analysis revealed the existence of 2 distinct clusters, with 1 cluster exhibiting a higher expression of proliferation-related genes and the other cluster enriched in pigmentation-related genes confirming that the pigmentation and proliferation programs are indeed maintained in distinct melanocyte subpopulations in vivo (Fig 1B and 1C).

To address the remaining questions, we revisited the NHEMs in culture. We used an already established mode of altering melanocyte phenotype using forskolin (fsk) as an external agent. fsk elicits a UV-like response in melanocytes by activating cyclic adenosine monophosphate (cAMP) and the downstream signalling cascade. Daily treatment of pigmented MNT1 cells results in a more proliferative phenotype, while treatment every alternate day gives rise to a hyperpigmentation response at the population level [7]. To dissect these phenotypes at a

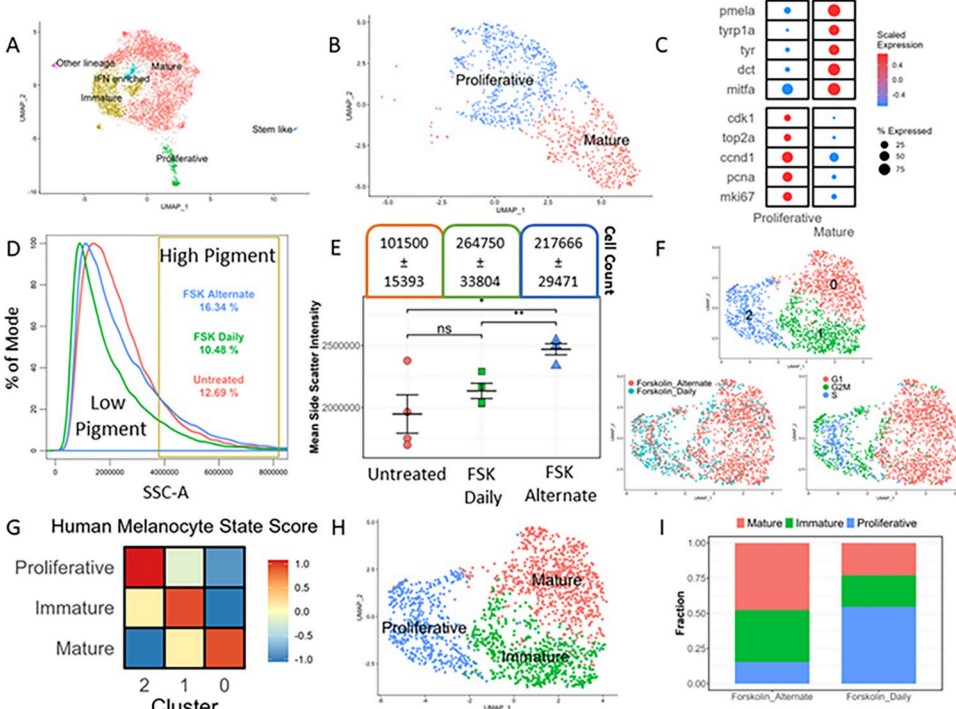

**Fig 1. Pigmentation and proliferation programs are maintained in distinct melanocyte subpopulations.** (A) UMAP visualisation of the human skin derived melanocyte coloured by the identified states that are labelled. This scRNA-seq data is meta-analysed from non-cultured NHEMs from skin (GSE151091). (B) UMAP visualisation of the zebrafish melanophores (mitfa+ cells isolated at 24 hpf) coloured by the identified states. This scRNA-seq data is submitted to (GSE240655). (C) Dot plot depicting markers of proliferative and mature states of the zebrafish melanophores (mitfa+ cells isolated at 24 hpf) with the bubble size showing the percent of cells expressing the gene and colour showing the scaled mean expression value (Wilcoxon–Mann–Whitney test with average log fold change >0.25 and adjusted $p$-value ≤0.05). (D) Density plot showing the distribution of pigmentation in cultured NHEM with daily or alternate day treatment of forskolin. (E) Mean side scatter intensity and cell count of cultured NHEM treated with daily or alternate day treatment of forskolin (unpaired Students $t$ test, $n = 4$, *: $p$-value ≤0.05, **: $p$-value ≤0.01, ns: $p$-value >0.05). (F) UMAP visualisation of cultured NHEM coloured by the clusters (top), treatment (bottom left), cell cycle phase (bottom right). This scRNA-seq data is submitted to (GSE233137). (G) Heatmap showing the gene-set activity scores (z-score of mean) of each of the cultured NHEM clusters for the human melanocyte mature, immature, and proliferative state. (H) UMAP visualisation of cultured NHEM coloured by the annotated states based on cluster markers and human skin melanocyte state score. (I) Stacked bar plot showing the proportion of different melanocyte states upon daily and alternate forskolin treatment. All numerical data are listed in S1 Table and S1 Data. hpf, hours post fertilisation; NHEM, normal human epidermal melanocyte.

more granular level, we resorted to a flow cytometry-based approach to estimate pigmentation at a single-cell resolution [29], while cell count served as a surrogate for proliferation. In the untreated melanocytes, we observed a gradient of pigmentation with a non-normal distribution containing a shoulder peak representing highly pigmented cells (Fig 1D). While fsk treatment increases the melanocyte population compared to untreated, the change observed in alternate day treated cells are higher compared to the daily treatment of fsk (Fig 1E). This suggests that fsk effects are retained in NHEMs and hence this model offers a platform to dissect these potential functional states.

We performed single-cell RNA sequencing on the daily and alternate-day fsk-treated melanocytes. Upon dimensionality reduction and clustering, we observed 3 clusters, with 2 (clusters 0 and 1) composed of cells mostly from the alternate day-treated sample while the third (cluster 2) contained cells mostly from the daily-treated sample (Fig 1F). Scoring these clusters for the cell cycle genes revealed that clusters 0 and 1 primarily consisted of cells in G1 phase

(non-cycling) while cluster 2 consisted of cells in S/G2/M phase (cycling) (Fig 1F). To ascertain the identities of these clusters, we scored the cells for human skin melanocyte states. Cluster 0 scored highest for the mature state, cluster 1 for the immature state, and cluster 2 for the proliferative state (Fig 1G). This, along with the cluster markers and GO term enrichment, allowed us to infer the identity of the clusters (Fig 1H). Based on the cluster composition, the daily and alternate day-treated samples contained all 3 identified states in different proportions. Although the stem cell-like and IFN-enriched states were not observed, probably due to non-conducive culture conditions, 3 of the 5 states namely proliferative, mature, and immature were faithfully recapitulated in cultured NHEMs. Our observations thus confirmed that, indeed, these melanocyte states are stable and coexist. As inferred from the flow cytometry data, alternate day treatment enriches the mature states, whereas daily fsk treatment enriches the proliferative state (Fig 1I). This observation suggests that the distinct coexisting states of melanocytes identified in this study are also interconvertible, depending upon the external cue.

However, to confirm the existence of these states beyond RNA-level information obtained from single-cell RNA sequencing, we conducted immunofluorescence (IF) analysis using 2 distinct markers for proliferation and pigmentation: Ki67 and Tyr, respectively. Normal human epidermal melanocytes (NHEM) treated alternately or daily with fsk were subjected to IF. Ki67 labelled a small subset of cells under untreated conditions, and this proportion increased with daily fsk treatment compared to alternate day treatment (S3A and S3B Fig). The presence of these 2 states was further validated through western blot analysis using cMyc as a proliferation marker and Tyr for differentiation (S3C Fig). This was also corroborated with MNT-1 cells using both IF as well as FACS analysis (S4 Fig). Whether these 2 states are equally responsive to the cue and how these transitions are orchestrated remains to be understood.

## Low and high pigment states correspond to proliferative and mature transcriptional states in B16 progressive pigmentation model

To identify the core networks maintaining the melanocyte cell state dynamics, we leveraged the already established B16 pigmentation model that mimics the tanning and de-tanning response [30,31]. This model offers a temporal resolution of pigmentation induction, allowing us to capture any alterations in cell states and complement the NHEM-based approach. Progressive pigmentation of the cell population is observed at the bulk level in this system. However, heterogeneity in melanin content is observed at the microscopic level. Flow cytometry offers a way to identify pigmented cells within the population [29,32,33]. Henceforth, we utilise flow cytometry-based estimation of side scatter as a surrogate measure for pigment content in melanocytes. We used this information to develop a method to estimate melanin content in single cells using imaging flow cytometry (S5A–S5C Fig). Though at day 7, the cell pellet is completely black with eumelanin, imaging flow cytometry revealed that the cells show a clear bimodal pigmentation distribution suggesting the presence of broadly 2 kinds of population: low pigment (LP) and high pigment (HP) (S5D Fig). This confirms the presence of a heterogeneous cell population in this progressive pigmentation model, endorsing the observations made in cultured NHEMs and melanocytes in vivo.

To map the transcriptional footprint of the low- and high-pigmentation states at day 7, we sorted these 2 populations using a flow cytometer (S5E Fig) and subjected them to single-cell RNA sequencing. Dimensionality reduction and unsupervised clustering revealed that low- and high-pigmented cells segregated into 2 different clusters. Interestingly, upon increasing the resolution further, 2 distinct subclusters emerged within the low- and high-pigment

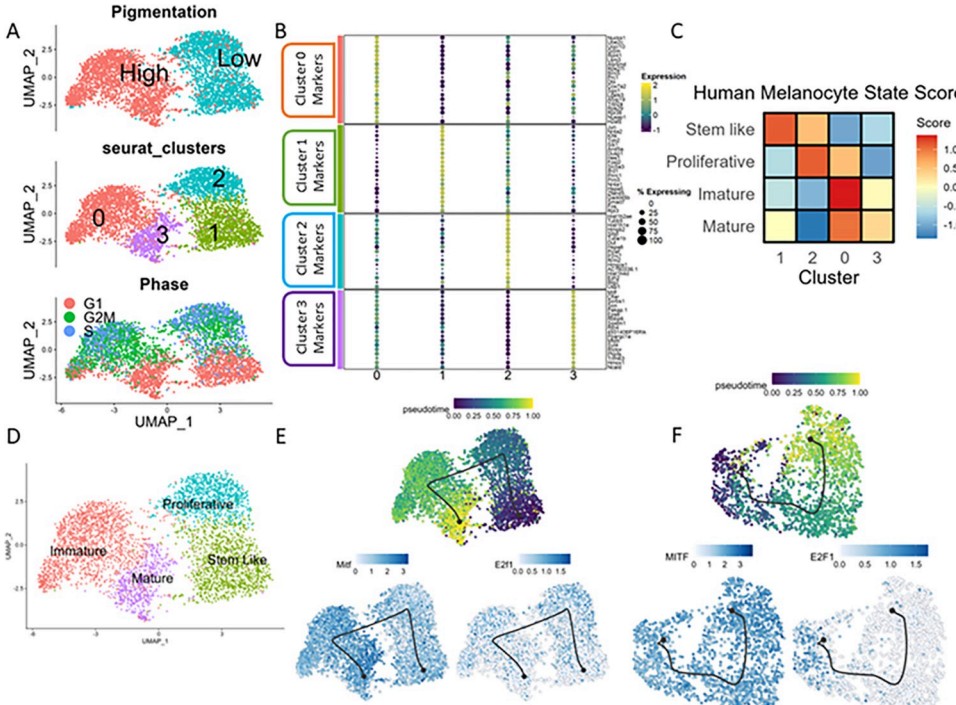

**Fig 2. Correlation of pigmentation with proliferative and mature states in B16 cells.** (A) UMAP plot showing B16 day 7 cells coloured by pigmentation (top), seurat clusters (middle), or cell cycle status (bottom). This scRNA-seq data is submitted to GSE233136. (B) Dot plot showing the top 20 marker genes enriched in each cluster with the size showing the percent of cell expressing the gene and colour showing the scaled mean expression value in each cluster (Wilcoxon–Mann–Whitney test with average log fold change >0.25 and adjusted *p*-value ≤0.05). (C) Heatmap showing the gene-set activity scores (z-score of mean) of each of the B16 clusters for the human melanocyte mature, immature, proliferative, and stem cell-like state. (D) UMAP plot showing B16 day 7 cells coloured by the cell states identified using marker genes, GO term enrichment analysis, and scores for human melanocyte cell states. (E) UMAP plot showing B16 day 7 cells coloured by pseudotime (top left), Mitf expression (bottom left), or E2f1 expression (bottom right). (F) UMAP plot showing cultured NHEM coloured by pseudotime (top left), MITF expression (bottom left), or E2F1 expression (bottom right). This scRNA-seq data is submitted to GSE233137. All numerical data are listed in S1 Table and S1 Data. NHEM, normal human epidermal melanocyte.

clusters (Fig 2A). Scoring the cells for the cell cycle genes showed that both the low- and high-pigmented clusters contained a G1 (non-cycling) and an S/G2/M (cycling) subpopulation (Fig 2A). Cluster markers and their GO term enrichment allowed us to infer the possible functional state of each cluster. Cluster 3 was enriched for mature state markers such as *Mitf* and *Tyr*, while cluster 2 was enriched for proliferative markers such as *Mki67* and *Top2a*. More interesting was cluster 1, containing stem cell-like markers *Klf4* and *Hes1*, but was relatively less proliferative than the other clusters suggesting a quiescent state. Cluster 0 had enrichment of both mature and proliferative markers, possibly representing an intermediate state before the melanocytes acquire a hyperpigmented mature state (Fig 2B). To confirm the identity of each cluster, we scored each of these for the previously identified human melanocyte state markers. Scoring confirmed our inference based on the cluster markers, with cluster 1 having the highest score for the human melanocyte stem cell-like state markers and cluster 0 having highest score for the immature state. Cluster 2 scored highest for the proliferative state, while cluster 3 for the mature state (Fig 2C). We scored these clusters for melanocyte states derived from mouse hair follicle single-cell RNA sequencing data [14] and obtained similar results (S2E Fig).

These results allowed us to annotate the B16 cell states (Fig 2D). We confirmed the existence of distinct cell states in B16 cells through 2 independent approaches: immunofluorescence and western blot-based analysis. For immunofluorescence, light and heavily pigmented colonies were imaged for the presence of ki67 and Tyr (S6A and S6B Fig). Further validation was performed using western blot analysis after FACS sorting the 2 populations based on side scatter (S6C Fig). While the markers were differentially enriched as anticipated in the western blot, it was interesting to observe that the highly proliferative LP colony had a high ki67 labelling with a concomitant low Tyr. Pigmented cells were low on ki67 and had high Tyr levels confirming the existence of 2 cell states as observed in the single-cell sequencing data.

The presence of similar cell states between the B16 mouse melanoma cells and NHEMs encouraged us to use the B16 pigmentation model to uncover the underlying gene regulatory networks (GRNs) governing melanocyte states. It was interesting to observe that all these states were maintained even at this stage in the pigmentation model when presumably the cells would have attained their terminal state of differentiation. Whether these states could switch from one to another was still unclear. To test this in silico, we used the Dynverse [34] package in R to infer a trajectory that would indicate the transition potential of these states. Upon applying the Scorpius algorithm, we observed a linear trajectory starting from the stem cell-like state and ending at the mature state with the proliferative and the immature states in between (Fig 2E). This highlights the switching potential of the stem cell-like state to the mature hyperpigmenting state if the need arises. A similar inference could be derived from cultured NHEM single-cell data wherein the cells transit from a proliferative to a mature state via the immature state in between (Fig 2F). This explains the different proportions of cell states observed with fsk treatments. Thereby, using 2 different cultured cell-based models, we demonstrate the coexistence of multiple melanocyte states with the possibility of transitions between these states.

## Progressive pigmentation model exhibits stochastic melanocyte state transitions

NHEM-based state transitions are unlikely to be driven by genetic mutations. The progressive pigmentation model, being a cancer cell-based system, could have acquired mutations over the course of time, resulting in the observed heterogeneity. The other possibility would be that at some point during progressive pigmentation, a few cells "switch on" the expression of pigmentation genes and then are destined to become pigmented, which would suggest stochastic (nongenetic) state transitions. To rule out the genetic component, we applied the Luria–Delbrück's fluctuation test and used a single clonal cell-derived population for further experiments. Two independent clonal populations gave rise to the similar bimodal distribution observed earlier, suggesting this phenomenon to be nongenetic in nature.

Since this progressive pigmentation model is a cell density-dependent phenomenon, we slightly modified the Luria–Delbrück's approach to test the genetic or nongenetic nature of this heterogeneity. Herein, we subjected the cells to one round of pigmentation and different treatments to obtain varying proportions of low- and high-pigmented cells (Fig 3A). We then started another round of pigmentation process using these differentially pigmented cells. The pigmentation state of the cells at the end of the second round of pigmentation was assessed using imaging FACS. We observed that irrespective of the starting proportion of pigmented cells, the final distribution of pigmentation was the same for all samples (Fig 3B). Since, in this model, cells grow as separate colonies, we assessed the colony pigmentation using high-throughput imaging. We observed that the colonies also showed the same pigmentation distribution irrespective of the starting proportion of pigmented cells (S7A and S7B Fig), thereby confirming that the transitions in this system are stochastic in nature.

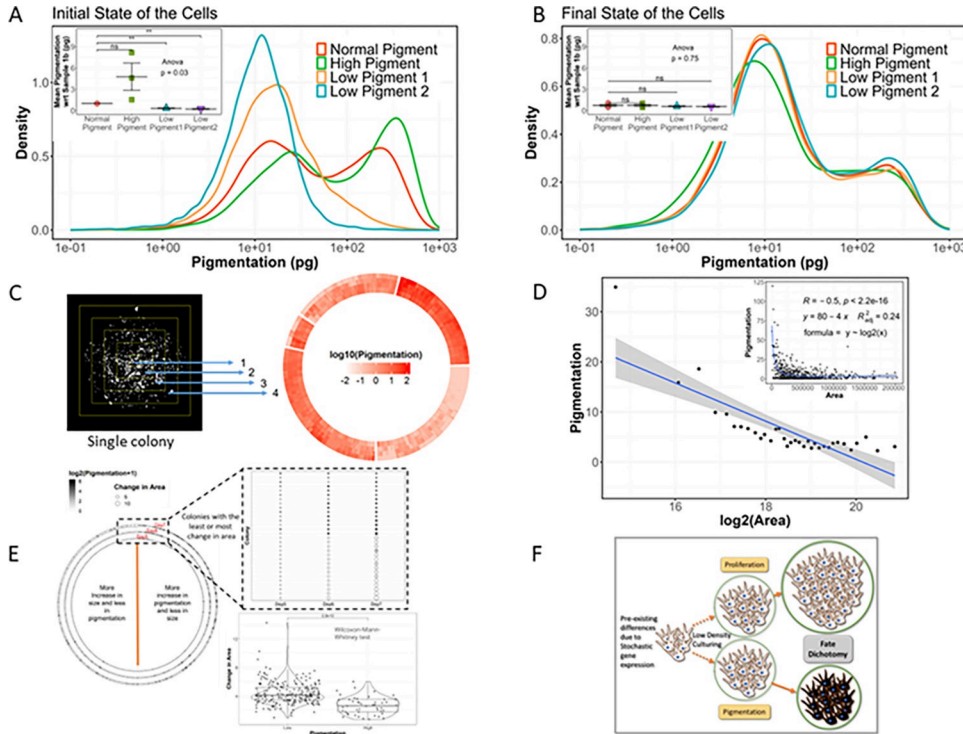

**Fig 3. Progressive pigmentation model exhibits stochastic melanocyte state transitions.** (A) Density plot showing the distribution of pigmentation (log scale) in single cells across differentially pigmented B16 samples with normal, high, or low pigment content achieved by treating with iso-butyl methyl xanthine (IBMX) or phenyl thiourea (PTU), respectively. Inset: statistical analysis of mean pigmentation normalised to normal pigmented sample, Student's *t* test, $n = 3$, **: *p*-value $\leq 0.01$, ns: *p*-value $>0.05$. Normal pigment: untreated, high pigment: 50 μm IBMX, low pigment 1 3b: 100 μm PTU, low pigment 2: 200 μm PTU. (B) Density plot showing the distribution of pigmentation (log scale) in single cells at the end of the second round of pigmentation model setup using differentially pigmented B16 samples in panel A. Inset: statistical analysis of mean pigmentation normalised to normal pigment sample in panel A, Student's *t* test, $n = 3$, **: *p*-value $\leq 0.01$, ns: *p*-value $>0.05$. (C) Representative inverted image of a single B16 colony at the end of pigmentation (left panel). Colonies were divided into concentric rectangles and the pigmentation in each sectors quantified. The circular heatmap (right) shows the pigmentation in each of the 4 sectors (inner to outer circle) for each colony. (D) Binned scatter plot showing the inverse relation between colony area and pigmentation. Inset: Raw data scatter plot with correlation coefficient derived from Pearson's correlation analysis and the equation derived from linear regression analysis between pigmentation and log2(Area). (E) (Left) Time-course tracking and analysis of colony pigmentation (colour of the circle) and area (size of the circle) from days 5 to 7 of the progressive pigmentation model. (Top right) Dot plot showing the top colonies with least change in pigmentation or highest change in area. (Bottom right) Statistical analysis of change in area from days 5 to 7 between low- and high-pigmented colonies (Wilcoxon–Mann–Whitney test). (F) Schematic representation of the dichotomy in fate decision resulting in 2 broad states in the progressive pigmentation model. All numerical data are listed in S1 Data.

Interestingly, the median standard deviation in mean grey value for intra-colony variability (2.1) was lower than the inter-colony variability (9.9) (Fig 3C). This pigmentation distribution of colonies suggested that the cells maintained 2 broad states with either low or very high pigmentation (S7C Fig). Since each colony is derived from a single cell, this suggests that the cells are indeed biased towards maintaining either of the 2 broad phenotypic states of proliferation or pigmentation. Microscopically, we observed that the colonies in the low-pigmented state were bigger compared to the high-pigmented colonies. Pearson's correlation analysis between the colony pigmentation and area was −0.5 suggesting a moderate negative correlation between pigmentation and colony area (Fig 3D). This is indicative of pigmenting cells to have a lower proliferation potential.

As pigmentation is a time-dependent slow process, it takes around 4 to 7 days before the actual phenotype presents itself and is detectable in the cell. With this, it becomes important to estimate the stage at which the pigmentation process starts, which presumably would be earlier than the phenotype is observable. A time-resolved model such as this progressive pigmentation confers an advantage to simultaneously infer the kinetics of both pigmentation and proliferation. We performed high-throughput imaging of the colonies on days 5, 6, and 7 of the pigmentation model. We observed that pigmented colonies on day 5 maintained their high-pigmentation state till day 7 and concomitantly had a lesser increase in colony area. Whereas less-pigmented colonies on day 5 proliferated more, maintaining their low pigmentation status till day 7. Thereby, these cells commit towards either pigmentation or proliferation fate as early as day 5 in this system, suggesting an early fate biasing decision that is retained in the daughter cells (Fig 3E).

These analyses along with the scRNA seq revealed that B16 cells, when put through the progressive pigmentation exhibit stochastic state transitions leading to 2 broad phenotypic states: low pigmentation (LP) coupled to high proliferation and high pigmentation (HP) coupled to low proliferation (Fig 3F).

## Stochastic gene expression leads to transcriptionally distinct melanocyte cell states

Visually the phenotypic heterogeneity in the pigmentation model was evident. From the time course tracking of individual colonies, it could be inferred that once the cells commit to a particular fate of either pigmentation or proliferation, their progeny retains the same state. Having ruled out the genetic factor, it is likely that epigenetic inheritance could explain the daughter cells retaining the parental state [17,35]. To address both the transcriptomic footprint and the epigenetic landscape, we performed single-cell multi-omics sequencing across days 0, 3, and 5 of the progressive pigmentation model with varying levels of pigmentation (S7D Fig).

Quality control was performed based on both RNA and ATAC parameters, and the RNA data was analysed first to identify the transcriptionally distinct clusters. We observed 4 different clusters possibly representing 4 different states. Colouring the UMAP based on the day revealed that 3 of the 4 clusters were predominantly composed of cells pertaining to different days of the pigmentation model. A technical replicate of day 5 was included in the analysis and both the technical replicates overlayed completely, confirming the absence of batch effects driving the clustering. We then used the weighted nearest neighbour approach in Seurat and Signac [36] to make a joint neighbourhood graph representing both the RNA and ATAC data in the UMAP space. Although the ATAC data alone was able to explain the differences in the days 0, 3, and 5 cells, it was not enough to segregate cluster 3 from the rest. Therefore, we used the RNA data to annotate the clusters. Based on the differentially expressed genes in each cluster and GO term enrichment (S1 Table), we annotated the clusters as B16 native state enriched in both stem cell-like (*Sox10*) and proliferative signatures (*Cenpa*, *Cdk1*), differentiating state enriched with markers such as *Mitf*, *Oca2*, and *Mlph*, proliferative state with high expression of *Mki67* and *Top2a* and a mature state having an enrichment of classical melanin synthesis genes such as *Tyr* and *Dct* among others (Fig 4A). We observed that the native, differentiating, and proliferative states were mainly present at days 0, 3, and 5, respectively, while the mature state emerged as early as day 3 (Fig 4B). This was surprising because visually pigmented cells could not be detected on day 3.

Enrichment of different state markers in the day 0 cluster indicated the preexistence of heterogeneity in the starting population. To investigate this further, we subclustered the day 0 sample individually. We used the markers of all the other states in the pigmentation model as

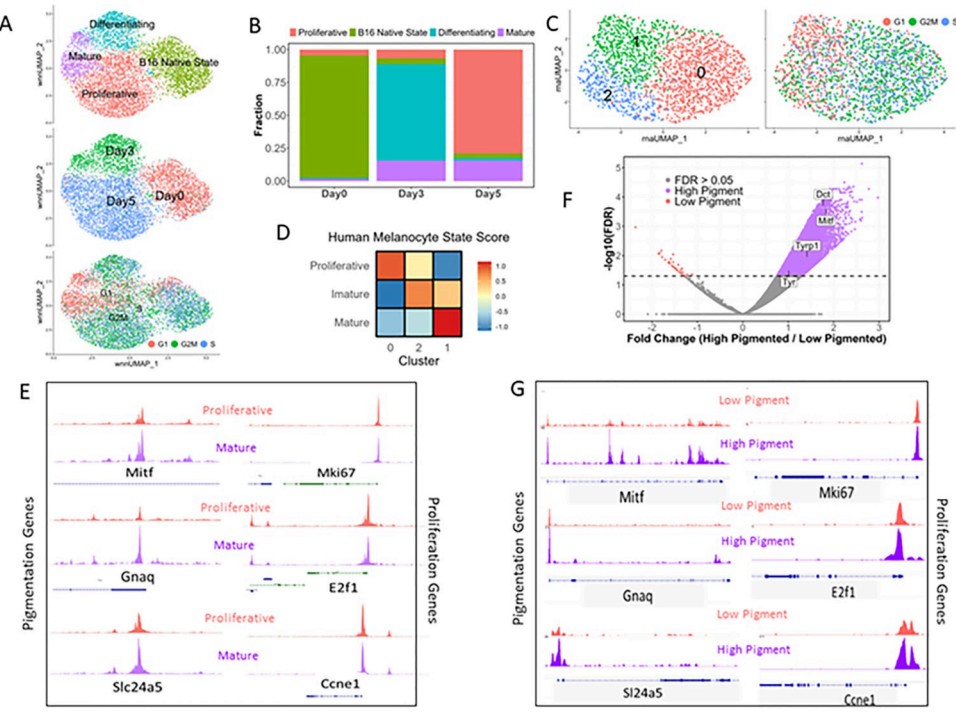

**Fig 4. Stochastic gene expression leads to transcriptionally distinct melanocyte cell states.** (A) UMAP plot showing B16 cells coloured either by the clusters (top), the day of the pigmentation model (middle), or by their cell cycle status (bottom). This multiome data is submitted to GSE233134. (B) Stacked bar plot showing the distribution of the 4 identified states at different days of the progressive pigmentation model. (C) UMAP plot showing subclusters of day 0 (initial state) of B16 cells coloured either by the clusters (left) or by their cell cycle status (right). (D) Heatmap showing the gene-set activity scores (z-score of mean) of each subcluster of day 0 B16 cells for the human melanocyte mature, immature, and proliferative state. (E) DARs derived from the single cell ATAC seq data around pigmentation genes (significantly different, adjusted $p$-value ≤0.05) and proliferation genes (not significant, adjusted $p$-value >0.05) between mature and proliferative states. This ATAC seq data is submitted as a part of multiome to GSE233134. (F) Volcano plot of H3K27Ac ChIP-seq data from low and high pigmentation cells, showing genes whose regions are differentially acetylated. Some of the key pigmentation genes are highlighted. The Chip-Seq data is submitted to GSE233135. (G) DHAcRs based on H3K27Ac ChIP-seq data around pigmentation genes (significantly different, adjusted $p$-value ≤0.05) and proliferation genes (not significant, adjusted $p$-value >0.05) between mature (high pigment) and proliferative states (low pigment). All numerical data are listed in S1 Table and S1 Data. DAR, differentially accessible region; DHAcR, differentially hyperacetylated region.

variable features for dimensionality reduction and clustering. The cells segregated into 3 clusters with cluster 0 having relatively high expression of proliferative state markers and cluster 1 having enrichment of mature state markers (Figs 4C and S7E). Cell state scoring for the human melanocyte states confirmed the relative enrichment of proliferative, mature, and immature state markers in clusters 0, 1, and 2, respectively (Fig 4D). This confirmed our hypothesis that day 0 cells, even though a genetically and phenotypically identical clonal population, natively harbour a heterogeneous cell population, possibly due to stochastic gene expression. This heterogeneity is likely to prime the cells to switch into the proliferative or the pigmented state that are observed in the model.

From this analysis, we confirmed that during progressive pigmentation, cells start to differentiate by day 3 and give rise to 2 broad states: proliferative and mature at the end of day 5. These subpopulations correspond to the low- and high-pigmented populations observed on day 7 using scRNA seq as well as imaging-based techniques.

## Active enhancers guide the melanocytes towards the mature state

While the cultured NHEM model revealed the coexistence and the transition of the melanocyte states, the progressive nature of the B16 pigmentation model allowed us to trace the dynamics of these state transitions in a time-resolved manner. The time-course tracking of colonies revealed that the daughter cells retain the parental states of low- and high-pigmentation, pointing towards an epigenetic driving force. Epigenetics influence the cell state dynamics and possibly regulates the cell state heritability across multiple cell divisions [17,35,37,38]. To identify the key players governing these melanocyte states, we resorted to an integrated approach involving chromatin accessibility (scATAC seq), an appropriate upstream histone modification (ChIP seq) and the downstream gene expression footprint across the LP and HP population (scRNA seq).

We analysed the scATAC seq data of the single-cell multi-omics experiment performed on days 0, 3, and 5 of the pigmentation model. We observed enrichment of open chromatin regions around the TSS for all the 4 states (S7F Fig). We found subtle differences in chromatin accessibility with only 2,713 differentially accessible regions (DARs) (adjusted *p*-value ≤0.05) across the 4 states, with 371 regions enriched in the mature state (S1 Table). Melanocytes maintained 2 broad stable states, proliferative and mature, corresponding to the low- and high-pigment phenotype. We compared these 2 states for chromatin accessibility differences and observed 397 and 162 regions to be differentially accessible in these respective states (adjusted *p*-value ≤0.05). Mapping the differentially accessible regions to the nearby genes revealed a few regions associated with known pigmentation genes such as Mitf and Slc24a5. These regions had higher chromatin accessibility in the mature state while the regions associated with proliferation genes had comparable chromatin accessibility across both the states (Fig 4E). Motif enrichment analysis yielded several transcription factor motifs among these DARs. Since single-cell ATAC seq data is very sparse in nature, which may make it difficult to capture subtle differences in chromatin accessibility across substates of the same cell type, accounting for the fewer DARs observed.

H3K27ac has been shown to be abundant on pigmentation genes in melanocytes [7,39]. In one of our earlier studies, we have shown that, through a pH-mediated feed-forward loop, H3K27ac selectively activates pigmentation genes and accentuates melanocyte differentiation at the population level [40]. Hence, we decided to investigate whether H3K27ac differences could explain the existence of these cell states. As the epigenetic regulation is likely to precede the transcriptional differences observed on day 7, we selected day 5 for our analysis. We performed chromatin immunoprecipitation of H3K27ac followed by sequencing (ChIP seq) in LP and HP cells sorted using flow cytometry on day 5 of the pigmentation model (S7G Fig). Peak calling, using a well-established program, macs3 [41], gave us 13,008 peaks in the LP, while 25,999 peaks in the HP, a marked increase compared to LP. Annotating the peaks using the ChIPseeker package [42] in R revealed that around 80% of the marked regions were within the 3 kb promoter in LP, which reduced to around 50% in HP with a concomitant increase in the first intron and distal intergenic region (S8A Fig). Differential peak analysis using DiffBind [43] gave us 12,579 differentially marked regions (FDR ≤ 5%), with almost all of those (12,540) having increased H3K27ac in HP (Fig 4F). After associating these differentially marked regions with the nearby genes using the rGREAT package [44], we observed Mitf and some other pigmentation-associated genes to have higher H3K27ac in the HP population (Fig 4G). The regions associated with proliferation genes had similar H3K27ac levels across both states (Fig 4G) similar to what we observed in our scATAC data. Motif enrichment analysis using HOMER [45] revealed 242 motifs, corresponding to several transcription factors (TFs) including MITF and Lef1, in the differentially hyperacetylated regions (DHAcRs) of HP.

Thereby the mature state is primarily driven by histone acetylation possibly by influencing the chromatin accessibility. Whereas the proliferative state is not driven by histone acetylation and DARs do not prominently map to proliferative genes indicating alternate mechanisms of controlling the expression of key proliferation genes.

To decipher this, in the third approach we set out to identify TF activity that could explain the differential gene expression across LP and HP states on day 7. We used Dorothea and Viper libraries in R [46–48] to estimate the TF activity from the gene expression data. After constructing the TF activity by cell count matrix, we performed unsupervised clustering and observed that LP and HP segregated into 2 different clusters. This allowed us to identify the TFs that govern the differential transcriptome in the LP and HP population (S8B Fig). We then identified the common overlapping set of TFs whose footprint is represented in all the 3 independent analyses (scATAC, H3K27ac, and gene expression footprint) described above (Fig 5A). This resulted in a set of 7 TFs enriched in HP (Fig 5B). A similar analysis for LP population resulted in a larger set of 86 common TFs, as there were no DHAcRs in LP (S1 Table). Hence using DHAcR and DAR marks that overlap with TF enrichment data set, we decipher the orchestrator of pigmented state, whereas DAR overlap with TF enrichment was used for the proliferation state.

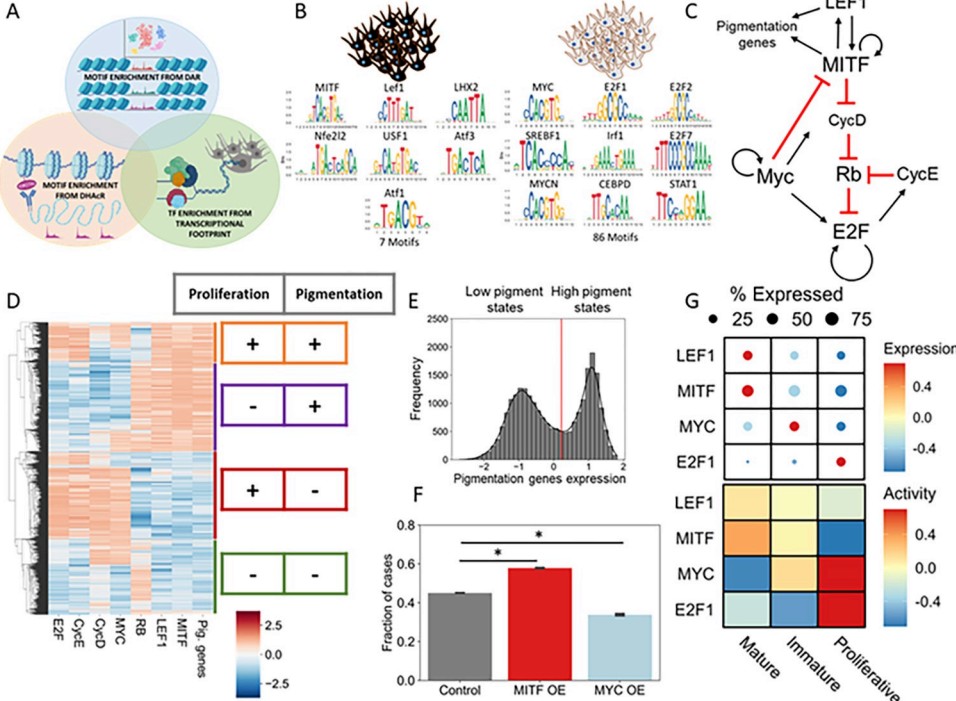

**Fig 5. Active enhancers guide GRN underlying melanocytes state transitions.** (A) Schematic showing combination of different approaches (scATAC, H3K27ac, and gene expression footprint) to identify melanocyte state-specific transcription factors. (B) Depiction of motifs derived from the intersection of motifs enriched in DHAcRs, DARs, and TF's enriched in melanocyte mature and proliferative states. (C) GRN, constructed from the common TF motifs, responsible for governing melanocyte cell state dynamics. (D) Steady-state solutions of simulations showing the presence of various stable states allowed by the GRN. Colour bar shows the z-normalised simulated expression levels of the listed genes with red representing higher expression and blue representing lower expression. (E) Histogram with corresponding kernel density estimate showing the bimodality in the expression of pigmentation genes based on simulations. (F) Simulated overexpression of MITF or MYC results in increase or decrease of high-pigment state, respectively (Students t test, *: p-value ≤0.01). (G) Heatmap showing expression (top) and activity (bottom) of TF's governing melanocyte cell states in cultured NHEM single-cell RNA seq data GSE233137. All numerical data are listed in S1 Table and S1 Data. DAR, differentially accessible region; DHAcR, differentially hyperacetylated region; GRN, gene regulatory network; NHEM, normal human epidermal melanocyte; TF, transcription factor.

## Melanocyte cell state GRN reliably predicts the state transitions

To construct a GRN that explains the coexistence and transitions of melanocyte cell states, we used the state-specific TFs identified above along with evidence from the literature [49–53]. Cell cycle as a bistable switch has been modelled previously in the context of general mammalian cell cycle entry decision-making circuit [54]. We used this information and constructed a GRN coupling pigmentation associated genes—MITF and LEF1 with the cell cycle associated genes—RB, E2F, and MYC TF along with 2 key cell cycle regulators CycE and CycD (Fig 5C).

Next, we sought to simulate to study the steady states allowed by this model and if it can explain the inverse association between melanocyte pigmentation and proliferation. To achieve this, we used the computational framework RACIPE (Random Circuit Perturbation), which solves a set of coupled ordinary differential equations (ODEs) to examine the various phenotypic states enabled by a GRN [55]. It does this by sampling an ensemble of kinetic parameter sets from a biologically relevant parameter range. For each distinct parameter set, it chooses initial conditions based on random sampling from within a log-uniform distribution for each node and then solves ODEs to obtain the possible steady states [56]. For some parameter sets, more than 1 steady state (phenotype) is achieved, suggesting possible stochastic switching among those states under the influence of noise.

Using RACIPE, we found that the constructed GRN can give rise to 4 main states—a high-pigmented state with low cell cycle genes, a state expressing both pigmentation and cell cycle genes, a high proliferation state with low pigmentation and a non-cycling stem cell-like state (Fig 5D). Simulated expression profile of pigmentation genes showed a clear bimodal distribution with distinct high and low pigmentation states (Fig 5E). Principal component analysis highlights the expression of MYC, E2F, and pigmentation genes (S8D Fig), unravelling 4 states similar to those observed in our single-cell data (Fig 2D).

Mitf is the master regulator of melanocyte biology, associated with multiple functions [57]. An increase in Mitf levels is suggested to control melanocyte differentiation and survival, and its expression relatively goes down with pigmentation [7,58]. Our multi-omics experiment suggested a similar scenario with the differentiating state (observed on day 3) showing the highest *Mitf* expression that decreased subsequently on day 5 (S8E Fig). One of the consistent themes that emerged from all our analyses was that in already differentiated melanocytes, *Mitf* had significantly high expression in the mature state compared to the proliferative state across all the data sets (S8F Fig). Therefore, we simulated the overexpression of MITF or MYC and investigated the effects on the frequency of high-pigmented state. While overexpression of MITF significantly increased the proportion of high-pigmented state, overexpression of MYC decreased it, thus pushing the cells towards a high-proliferative state (Fig 5F). Overall, the results suggest that the steady-state dynamics of the underlying GRN play a crucial role in the emergent property of inverse association between melanocyte pigmentation and proliferation programs.

## State switching in cultured NHEMs is guided by the melanocyte cell state GRN

We revisited the forskolin response in melanocytes that activates either proliferation or pigmentation programs based on the frequency of administration of fsk (Fig 1H). Our scRNA seq analysis demonstrated that the treatments rather altered the proportion of the cell states enriched in these 2 diametrically opposite programs within the population. The GRN derived from the B16 progressive pigmentation model explained the coexistence of distinct B16 cell states. We then tested whether this GRN could explain the switch in proliferation and pigmentation states in NHEMs with the addition of fsk. We derived the activity of the key TFs in this

GRN using the target gene expression in the proliferative and the 2 pigmenting states (mature and immature). We observed that LEF1 and MITF were active in the mature state, whereas MYC and E2F1 were active in the proliferative state (Fig 5G), suggesting that the same GRN is responsible for the coexistence and further orchestrates the transition of the human melanocyte states.

## Discussion

Here, using in vitro cell-based models and single-cell methodologies, we demonstrate that specified melanocytes exhibit distinct cell states that are governed by a GRN containing regulators of pigmentation (MITF, LEF1) and proliferation (MYC, E2F1). We coupled the known proliferation circuit with the pigmentation regulators and showed that variability in the expression/activity of these factors, under the influence of noise (stochasticity), resulted in the coexistence of multiple functionally distinct melanocyte states. Physiologically, during the hair cycle, melanocyte stem cells are known to differentiate to give rise to a mature pigmented state through a proliferative state. This is shown to be governed by the interplay of external signalling factors that are spatiotemporally controlled [14]. A recent study showed that the transit amplifying (proliferative) state can either differentiate into a mature pigmented state or dedifferentiate into an McSc state in a WNT-dependent manner [12]. Whether it is the cue that primarily induces these states or these are preexisting within the seemingly homogeneous melanocyte population remained unclear.

Stochasticity in gene expression has been shown to govern melanoma cell states, giving rise to a drug-resistant state even before the addition of any drug. These states are largely predetermined and are further strengthened by epigenetic programs resulting in clonal expansion [16–18]. In this current work, we observed preexisting transcriptional variability in day 0 cells with a mature-like and a proliferative subpopulation (Figs 4C, 4D, and S7E). It is likely that these cells are the ones that give rise to the mature state in the B16 progressive pigmentation model, and low-density culturing appears to be the trigger for these state transitions. Using high-throughput imaging on days 5, 6, and 7 in the model, we observed that the 2 broad states, LP and HP, are maintained throughout as independent colonies suggesting inheritance of the parental state across subsequent cell cycles. Our earlier work demonstrated the role of H3K27ac in melanocyte differentiation through a pH-mediated feed-forward loop involving Mitf and its targets. We observed hyperacetylation of pigmentation genes on day 5 [40]. In this work, we map the hyperacetylation of H3K27 predominantly to the HP state. The default activation state appears to be proliferation. This notion is strengthened by the observation that almost all proliferation related genes that are differentially expressed have comparable accessibility across the 2 states. This suggests that the mature and the proliferative state, much like the drug-resistant state in melanoma, are preexisting, with the mature state getting reinforced via hyperacetylation at H3K27 whenever a trigger is presented.

In a normal physiological scenario, melanocytes are constantly exposed to UV radiations resulting in the tanning response involving cAMP pathway, which in this study is simulated by fsk treatment. While the response of cells to fsk is well recognized, an elegant study by Malcov-Brog and colleagues [7] demonstrated that proliferation and pigmentation are differentially activated depending upon the frequency of exposure. Our study, using in vitro cell models, deciphers that melanocytes exhibit these 2 programs in separate subpopulations that preexist before the trigger. Differential exposure to fsk results in the enrichment of one state. On recurrent exposure, the proliferating state is expanded, while infrequent exposure leads to the enrichment of the pigmented state. Mitf dynamics remain at the centre of melanocyte functioning. Malcov-Brog and colleagues [7] demonstrated that an increase in Mitf levels induces

the cell survival program in the melanocyte population. In the B16 progressive pigmentation model, we observed the highest *Mitf* levels in the differentiating state, followed by a decrease, once the cells differentiate at day 5 (S8E Fig). Interestingly after differentiation, we observed that cells in a proliferative state had relatively lower *Mitf* expression compared to the mature population (S8F Fig). Drawing parallels from this model suggests that exposure of melanocytes to a cue such as UV radiation/fsk results in the elevation of Mitf, leading to the initiation of the cell survival program and increased proliferation. Subsequently, Mitf decreases, and in the absence of additional exposure, cells with relatively higher levels of Mitf get biased towards the mature state while others remain in the proliferative state. The proliferation regulators such as Myc then further suppress Mitf to maintain the other cells in the proliferative state. However, further investigation is needed to delineate these cell state dynamics in detail. In all, stochasticity coupled with a programmed regulatory network explains the coexistence of melanocyte cell states, and ultimately, the population-level diversity. The coexistence of these states is likely to permit the system to adapt quickly to external cues by altering the cell state dynamics.

## Material and methods

### Ethics statement

Fish experiments were performed in strict accordance with the institutional animal ethics approval (IAEC) of the CSIR-Institute of Genomics and Integrative Biology (IGIB), India (Proposal No 45a). IEAC approval number–IGIB/IAEC/25/28/202. All efforts were made to minimise animal suffering.

### B16 pigmentation model and NHEM culturing

All the cells were maintained at 37˚C and 5% $CO_2$.

B16 cells were cultured in Dulbecco's Modified Eagle's Media (DMEM-high glucose, Sigma-Aldrich, D5648) supplemented with 10% fetal bovine serum (FBS, Gibco, 10270106). MNT-1 cells were cultured in DMEM-high glucose (Sigma-Aldrich, D5648) supplemented with 20% FBS (Gibco, 10270106) and 10% AIM-V media (Gibco, 12055083).

For the progressive pigmentation model, B16 cells were seeded at a very low density of 100 cells/cm$^2$ and cultured for appropriate number of days as per the experimental requirement, for a maximum of 7 days. As per the requirement, the cells were either treated with 50 μm 3-Isobutyl-1-methylxanthine (IBMX, Sigma-Aldrich, I7018-100MG) to enhance pigmentation or with 100/200 μm N-Phenylthiourea (PTU, Sigma-Aldrich, P7629) to prevent pigmentation, 1 day after seeding the cells. The establishment of progressive pigmentation in B16 cells is available in the protocol.io [30].

NHEMs were purchased from Lonza (CC-2504). Cells were revived in Lonza MGM4 media (CC-3250) supplemented with supplied factors (CC-4435). Post revival, cells were transferred to and maintained in Medium 254 (Gibco M254500) supplemented with human melanocyte growth supplement (HMGS-2, Gibco S0165). A total of 50,000 cells were seeded in 3 wells of a 6-well plate. One well was treated daily while the other was treated every alternate day with 20 μm forskolin (Sigma-Aldrich, F3917) for 7 days. Equal volume of DMSO was added in the remaining well to serve as a control.

### Melanin estimation using NaOH method

Melanin estimation was performed as described earlier [59]. Briefly, equal number of cells (approximately 25 Lacs) from each sample were centrifuged at high speed, dissolved in 250 μl of 1N NaOH, vortexed, and kept at 80˚C for 2 h. The solution was spun down at high speed

and 100 µl supernatant, in duplicates, was transferred to a 96-well plate. The absorbance was measured at 400 nm. Synthetic melanin (Sigma-Aldrich, M8631) was used to generate the standard curve and estimate the melanin content in each sample.

## Melanin estimation using imaging flow cytometer

Cells were trypsinized, washed with 1× Dulbecco's phosphate buffer saline (DBPS) 2 times and resuspended in 1× DPBS. Each sample was run sequentially on Amnis Imagestream Mk II after initializing the system. Initial gating was done using the area and aspect ratio parameters to select only single round cells. A total of 10,000 cells were captured for each sample after the gating. The analysis was done on the Ideas software.

## Pigmentation estimation using ImageJ

All the microscopy images were analysed in Fiji. A threshold was applied to the images such that only the dark spots corresponding to melanin were visible in pigmented colonies and nothing was visible in depigmented colonies. Colonies were marked using the ROI tool and mean grey value estimation for each colony was taken as the pigmentation content in the colonies. Any colonies appearing to be merged (morphologically) from multiple individual colonies were left out of the analysis.

## Immunocytochemistry

Cells (B16 or MNT-1 or NHEMs) were seeded on UV-treated sterile coverslips (Corning) in 6-well plates. Cells were fixed with 4% paraformaldehyde in PBS for 10 min at 37˚C. Further, they were permeabilized with 0.1% Triton X-100 in PBS for 15 min at room temperature on slow orbital shaking. After 3 consecutive washes with 1× PBS, cells were blocked with 5% NGS (Jackson's immunoresearch) for 1 h at room temperature on slow orbital shaking. Cells were given a single wash of PBS and incubated with anti-Ki67 antibody (ab16667, 1:100) or anti-TYR (custom synthesised genscript, 1:100) in 1% NGS for 2 h at room temperature. Post primary antibody incubation cells were washed thrice with PBST (PBS + 0.1% Triton X-100) for 5 min each on slow orbital rotation and then incubated in secondary Alexa Fluor 488 anti-rabbit antibody (1:500 in 1% NGS) for 1 h at room temperature in dark. Finally, cells were washed thrice with IX PBST and mounted on glass slides with 10 µl of antifade DAPI and sealed the coverslips using acetone. Imaging was done using Leica confocal microscopy and quantitated using Image J software.

## Western blot analysis

Whole-cell lysates of B16 or MNT-1 or NHEMs were resuspended in NP40 Lysis Buffer (Thermo; ALF-J60766-AP) reconstituted with protease and phosphatase inhibitors (750 µl of NP40 lysis buffer, 1× PIC, 10 mM sodium pyrophosphate, 1 mM sodium orthovanadate, 10 mM sodium glycerophosphate, 1 mM phenyl methyl sulphonyl fluoride (PMSF). Cells were incubated in lysis buffer overnight at −80˚C and centrifuged at 13,000 rpm for 20 min at 4˚C, protein supernatant was collected and estimated using standard BCA protocol (Pierce BCA protein assay kit; Thermo). The 25 µg of the protein was used for proteins in the size range of 20 to 150 kDa. Primary antibody incubations were done for overnight in cold room (4˚C). Subsequent to incubation with the primary antibody, the membrane was washed thrice with TBST containing 0.1% tween and were further incubated with corresponding HRP conjugated secondary antibody at 1:10,000 dilution in 5% Skim milk or for 1 h at room temperature.

Membrane was washed in TBST with 0.1% Tween-20, thrice for 15 min each and developed using ECL reagent.

## Time-course tracking and analysis of B16 colonies

High-content live cell imaging was done at days 5, 6, and 7 of the pigmentation model. First the colonies at day 5 were marked using the ROI tool and then the same colonies were traced at day 6 and then at day 7 by using the edges of the well as reference. Colonies that got merged and were untraceable in the subsequent days were filtered out. Colony pigmentation and area was estimated using Fiji and the data was plotted in R Studio using ggplot2.

## Human melanocyte single-cell data analysis

The human melanocyte single-cell RNA sequencing data was analysed using Seurat v4 in R Studio. Raw counts were downloaded from GSE151091. The data was filtered for low-quality cells and gene count was normalised. Clustering and dimensionality reduction was performed using first 29 PC's. Melanocytes were extracted based on the gene expression profile and were further sub-clustered. The clusters were batch corrected using harmony.

## Single-cell RNA sequencing and analysis of B16 cells

B16 cells were taken at day 7 of the pigmentation model and sorted into low- and high-pigmented population using side scatter intensity. The single-cell RNA libraries were prepared for both samples using the 10× genomics Chromium Next GEM Single Cell 3′ Reagent Kit v3.1. The library QC and quantification was done using Agilent bioanalyzer HS DNA kit. The libraries were pooled and sequenced on the NextSeq 2000 platform. Raw bcl files were converted to final count matrix using Cell Ranger v6.1.2 software following the tutorial provided on the 10x genomics website. All the downstream analysis was performed in R Studio using Seurat v4 package. Low-quality cells and genes expressing in less than 3 cells were filtered out. The data was log normalised and 3,000 highly variable genes and 30 principal components were used for dimensional reduction and clustering.

## Single-cell multi-omics sequencing and analysis of B16 cells

B16 pigmentation model was setup on different days to obtain days 3 and 5 sample on the same day and the cells used to seed days 3 and 5 sample were maintained at high density and taken as day 0 sample. The multiome libraries were prepared using the Chromium Next GEM Single Cell Multiome ATAC + Gene Expression kit. The library QC and quantification was done using Agilent bioanalyzer HS DNA kit. The libraries were pooled and sequenced on the NextSeq 2000 platform. Raw bcl files were converted to final count matrix using Cell Ranger ARC v2.0.2 software following the tutorial provided on the 10x genomics website. All the downstream analysis was done in R Studio using Seurat v4 and Signac v1.9 packages. Both RNA and ATAC parameters were used for initial quality control. The clusters were identified using the RNA data and the ATAC data was overlayed on it using joint neighbour calling to finally arrive at the joint UMAP reduction.

## Melanocyte state score calculation

To score clusters for melanocyte states, the marker genes representing each state was obtained for human (GSE151091) and mouse (GSE147299) melanocytes, respectively. These state-specific marker genes were used as gene sets and gene set activity score was obtained for the clusters in the query data sets using AddModuleScore utility in Seurat v4. Then, the mean of the

scores were calculated for each state in each cluster and the z-scores of the mean values were plotted as heatmap.

## Single-cell sequencing and analysis of NHEMs

Forskolin daily and alternate treated cells were taken for library preparation using the 10x genomics Chromium Next GEM Single Cell 3′ Reagent Kit v3.1. The single-cell RNA libraries were prepared for both samples using the 10x genomics Chromium Next GEM Single Cell 3′ Reagent Kit v3.1. The library QC and quantification was done using Agilent bioanalyzer HS DNA kit. The libraries were pooled and sequenced on the NextSeq 2000 platform. Raw bcl files were converted to final count matrix using Cell Ranger v6.1.2 software following the tutorial provided on the 10x genomics website. All the downstream analysis was performed in R Studio using Seurat v4 package. Low-quality cells and genes expressing in less than 3 cells were filtered out. The data was log normalised and 3,000 highly variable genes and 22 principal components were used for dimensional reduction and clustering.

## Single-cell RNA sequencing and analysis of zebrafish melanophores

Tg(mitfa:GFP) line of the zebrafish were bred, raised, and maintained at 28.5˚C according to standard protocols [60] and were housed at the CSIR-IGIB, Mathura Road New Delhi, India. Embryos were staged based on time (hpf) and morphological features, according to [61]. Single-cell RNA sequencing was performed on 24 hpf mitfa:GFP positive zebrafish cells using 10x genomics Chromium Next GEM Single Cell 3′ Reagent Kit v3.1. Zebrafish embryos were processed and prepared for flow cytometry as described previously [62]. The cells positive for mitf:GFP were sorted using the BD FACSAria II cell sorter and taken up for library preparation according to manufacturer's instructions. Sequencing was performed on NextSeq 2000 platform. Raw bcl files were converted to final count matrix using Cell Ranger v6.1.2 software after generating the reference genome for zebrafish following the tutorial provided on the 10x genomics website. All the downstream analysis was performed in R Studio using Seurat v4 package. Low-quality cells and genes expressing in less than 3 cells were filtered out. The data was log normalised and 3,000 highly variable genes and 50 principal components were used for dimensional reduction and clustering. The cells expressing endogenous mitfa and representing the pigment lineage were extracted and clustered again using 24 principal components. The xanthophore cluster was removed and only the melanophore population was used for further analysis.

## H3K27ac chromatin immunoprecipitation and sequencing

Chromatin immunoprecipitation was performed using the protocol provided by Upstate Biotechnology with modifications as suggested in the Fast ChIP protocol. Briefly, B16 cells were taken at day 5 of the pigmentation model. Cells were sorted into low and high pigment population based on side scatter intensity, fixed with 10% formaldehyde (Sigma-Aldrich, F8775) at 37˚C for 10 min and resuspended in SDS lysis buffer (1% SDS (Sigma-Aldrich, L3771), 10 mM EDTA (Sigma-Aldrich, E6758), 50 mM TRIS (Sigma-Aldrich, T6066) (pH 8.1)) after washing with ice cold 1× DPBS containing protease inhibitors. Lysis was done for 30 min on ice and then chromatin lysate was obtained by manual shearing in a bath sonicator using 6 to 7 cycles of 7.5 min keeping the sonicator on for 30 s and off for 45 s. Chromatin was quantified using qubit HS DNA kit and equal amount of chromatin was taken for each set of replicates for chromatin immunoprecipitation using the anti-H3K27ac antibody (ab4729). Unsorted cells were used for control ChIP using anti-IgG antibody (ab172730). Libraries were prepared using NEB

Ultra II DNA library prep kit (E7103S) following manufacturer's instructions and sequencing was done, after pooling the libraries together, on the NextSeq 2000 platform.

## ChIP-seq analysis

All the analysis was done using default parameters unless specified. The raw bcl files were converted to fastq using bcl2fastq. Quality control was done using fastqc and bad quality bases were trimmed using trimmomatic (CROP:150 LEADING:3 TRAILING:3 SLIDINGWIN-DOW:4:15 MINLEN:50). Poly-G/poly-x overrepresented sequences and any remaining adapter traces were removed using fastp. Alignment was done to mm10 genome using bowtie2 and samtools was used to fix mate coordinates, sorting and marking duplicates to get the final bam file. MACS3 was used to call narrow peaks and bedtools was used to identify the common and unique peaks between the samples. Differentially acetylated regions were identified using DiffBind and rGreat was used to annotate the regions to the nearby genes in R Studio. HOMER was used for motif enrichment analysis in the differentially acetylated regions.

## Gene regulatory network simulations

Random Circuit Perturbation (RACIPE) is a computational framework that generates an ensemble of kinetic models for a given GRN and simulates its dynamics for a range of biologically relevant parameters and initial conditions. The RACIPE input network consists of inhibitory and activating links between each node. To calculate the expression of each node in the network, RACIPE uses a set of ODEs defined as follows: $dX_i/dt = gX_i\prod_j H_s(X_j, X_{ji0}, n_{ji}, \lambda_{ji}) - kX_i X_i$.

In the RACIPE framework, each node in the input GRN is represented by a concentration variable $X_i$, where $i \in \{1,2,3,4,5\}$. The expression level of each node is determined by a set of ODEs that consider various parameters, including the basal production rate g, basal degradation rate k, and the shifted Hill function $H_s$. The shifted Hill function is used to incorporate the activating or inhibitory links between nodes and to determine the production rate for each node. The parameters λ, n, and X0 correspond to each regulatory link in the network and represent the fold-change parameter, Hill's coefficient, and threshold value for the Hill's function, respectively. Through the RACIPE framework, an ensemble of kinetic models is generated for the given GRN, and its dynamics are simulated for a range of biologically relevant parameters and initial conditions. Initial condition for each node is randomly sampled from a log-uniform distribution of minimum to maximum levels of that node. RACIPE steady states obtained are z-normalised for all further analysis. The perturbation experiments were done by increasing the basal production rates by 20-fold and comparing the percent of high-pigmented states (z-score >0.22 obtained from the bimodal pigmentation score distribution for the control case) to the control case where the basal production rate was not over expressed.

## Statistical analysis and graphs

All the statistical analysis was performed in R Studio using ggpubr library. All the plots were made using ggplot2 library. *P*-value (P) >0.05 is represented as ns, $P \leq 0.05$ as *, $P \leq 0.01$ as **, $P \leq 0.001$ as ***, and $P \leq 0.0001$ as ****.

## Supporting information

**S1 Fig. Analysis of epidermal NHEM scRNA-seq data taken from GSE151091 (related to Fig 1).** (A) UMAP visualisation of the human skin melanocyte coloured by age group (top), donor ID (bottom left), plate ID (bottom right). (B) UMAP visualisation of the human skin melanocyte coloured by the clusters. (C) Dot plot showing the top 10 marker genes enriched

in each cluster, with size depicting the percent cells expressing the gene and colour depicting the scaled mean expression value in each cluster (Wilcoxon–Mann–Whitney test with average log fold change >0.25 and adjusted $p$-value ≤0.05). (D) Dot plot depicting top 5 GO terms enriched in each cluster identified using cluster markers. (E) Stacked bar plot showing the distribution of different age groups in each of the identified cell states. All numerical data are listed in S1 Data.
(DOCX)

**S2 Fig. Analysis of epidermal NHEM scRNA-seq data taken from EGAS00001002927 and mouse hair follicle data set (GSE147299) (related to Figs 1 and 4, respectively).** (A) UMAP visualisation of the human skin melanocyte coloured by the clusters. (B) Dot plot of depicting the top 10 marker genes enriched in each cluster with size showing the percent cells expressing the gene and colour depicting the scaled mean expression value in each cluster (Wilcoxon–Mann–Whitney test with average log fold change > 0.25 and adjusted $p$-value ≤0.05). (C) Dot plot depicting top 5 GO terms enriched in each cluster identified using cluster markers. (D) Heat map depicting the gene-set activity scores (z-score of mean) of each of the epidermal NHEM clusters for the human melanocyte mature, proliferative, IFN enriched, and stem-like state. (E) Heatmap depicting the scores (z-score of mean) of each of the B16 clusters for the mouse hair follicle melanocyte mature, proliferative and stem-like state. All numerical data are listed in S1 Data.
(DOCX)

**S3 Fig. Immunofluorescence and western blot analysis of daily and alternate day forskolin-treated NHEMs.** (A) Immunofluorescence images of NHEMs treated either daily or every alternate day with forskolin and stained for proliferative and pigmenting marker proteins. Nuclear DNA stained with DAPI (blue), Ki67 and TYR in green. Scale bars represent 150 μm. (B) Quantitation of corrected total cell fluorescence (CTCF) of individual cells for Ki67 and TYR in daily and alternate day forskolin-treated NHEMs. $p$-value via an unpaired, two-tailed Student's $t$ test, with significant values ($p < 0.05$) is displayed on the graph. (C) Western blot analysis of daily and alternate day forskolin-treated NHEM cells. Numbers below the blot represents fold change *wrt* daily forskolin-treated cells. (DF) Daily forskolin, (AF) alternate forskolin-treated NHEMs.
(DOCX)

**S4 Fig. Immunofluorescence, western blot, and FACS analysis of daily and alternate day forskolin-treated human MNT-1 cells.** (A) Immunofluorescence images of MNT-1 cells treated either daily or every alternate day with forskolin and stained for proliferative and pigmenting marker proteins. Nuclear DNA stained with DAPI (blue), Ki67 and TYR in green. Scale bars represent 150 μm. Quantitation of corrected total cell fluorescence (CTCF) of individual cells for Ki67 and TYR in daily and alternate day forskolin-treated MNT-1 cells. $p$-value via an unpaired, two-tailed Student's $t$ test, with significant values ($p < 0.05$) is displayed on the graph. (B) Western blot analysis of c-MYC and TYR in daily and alternate day forskolin-treated MNT-1 cells. Numbers below the blot represents fold change *wrt* daily forskolin-treated cells. (C) Quantitation of percent Ki67/TYR positive cells from 2 biological replicates of MNT-1 with daily or alternate day forskolin treatment. Two-way ANOVA was performed. Adjusted $P$-values: ** $P$-value <0.001, *** $P$-value <0.0001, **** $P$-value <0.00001. (D) Quantitation of mean fluorescence intensity per cell from 2 biological replicates of MNT-1 with daily or alternate day forskolin treatment. Two-way ANOVA was performed. Adjusted $P$-values: ** $P$-value <0.001, *** $P$-value <0.0001, **** $P$-value <0.00001.
(DOCX)

**S5 Fig. Assessment of low and high pigmentation melanocyte states (related to Fig 2).** (A) Bright field images of day 7 B16 low and high pigmenting colonies. The bottom panel represents images of single cells sorted using imaging flow cytometry from the differentially pigmented day 7 sample. Scale bars are indicated. (B) Pearson's correlation analysis between the average melanin content per cell estimated using NaOH method and the different parameters of imaging flow cytometer. (C) Linear regression analysis of mean brightfield intensity of imaging flow cytometer and average melanin content per cell estimated using NaOH method. (D) Distribution of pigmentation in B16 cells at day 7 of the progressive pigmentation model. (E) Side scatter intensity distribution (representing pigmentation) of B16 cells at days 0 and 7 of the progressive pigmentation model. Rectangle gate represents high pigment cells. All numerical data are listed in S1 Data.
(DOCX)

**S6 Fig. Immunofluorescence and western blot-based analysis of differentially pigmented B16 mouse melanoma cells.** (A) Immunofluorescence images of day 7 B16 cells depicting low (LP) and high (HP) pigmenting colonies that were stained for proliferative and pigmenting marker proteins. Nuclear DNA stained with DAPI (blue), Ki67 (green), and TYR (red). Scale bars represent 150 μm. (B) Quantitation of corrected total cell fluorescence (CTCF) of individual cells for Ki67 and TYR in low and high pigmenting B16 cells. (MEL LOW) low melanin, (MEL HIGH) high melanin content based on visual inspection in bright field image. $p$-value via an unpaired, two-tailed Student's $t$ test, with significant values ($p < 0.05$) is displayed on the graph. (C) Western blot images and quantitation of protein levels of TWIST 1, C MYC, LEF 1, TYR, and DCT in B16 day 7 cells sorted based on pigmentation. Quantitation with respect to low pigmenting cells is depicted below the blot images. (LP) Low pigmenting, (HP) high pigmenting B16 cells sorted based on FACS using side scatter information. Experiments were performed in duplicates.
(DOCX)

**S7 Fig. Imaging based assessment of differentially pigmented B16 states and quality control of scMultiomics data (related to Figs 3 and 4).** (A) Density plot showing the distribution of pigmentation (log scale) in single cells across differentially pigmented B16 samples. Inset: statistical analysis of mean pigmentation, Wilcoxon–Mann–Whitney test, $n = 3$, ****: $p$-value $\leq 0.0001$, ns: $p$-value $>0.05$. High pigment: 50 μm IBMX, low pigment: 100 μm PTU. (B) Density plot showing the distribution of pigmentation (log scale) in single colonies at day 7 of the pigmentation model setup using the B16 samples in panel E. Inset: statistical analysis of mean pigmentation, Wilcoxon–Mann–Whitney test, ****: $p$-value $\leq 0.0001$, ns: $p$-value $>0.05$. (C) Distribution of pigmentation in B16 colonies at day 7 of the progressive pigmentation model. (D) Side scatter intensity distribution (representing pigmentation) of B16 cells at days 0, 3, and 5 of the progressive pigmentation model. Rectangle gate represents high pigment cells. (E) Dot plot showing the top 15 marker genes enriched in each subcluster of day 0 B16 cells with the size showing the percent of cell expressing the gene and colour showing the scaled mean expression value in each cluster (Wilcoxon–Mann–Whitney test with average log fold change $>0.2$ and adjusted $p$-value $\leq 0.05$). (F) Distribution of ATAC peaks around the TSS in different melanocyte states of the progressive pigmentation model. (G) Side scatter intensity distribution (representing pigmentation) of B16 cells at days 0 and 5 of the progressive pigmentation model. Rectangle gate represents high pigment cells. All numerical data are listed in S1 Data.
(DOCX)

**S8 Fig. Analysis of H3K27ac ChIP and scRNA-seq data sets (related to Figs 4 and 5).** (A) Plots showing distribution of H3K27ac peaks around the TSS (top), annotation of peaks (bottom left), and distribution of peaks relative to TSS in day 5 LP and HP population. (B) UMAP plot showing day 7 cells of the progressive pigmentation model coloured by clusters identified using TF activity (top) and pigmentation (bottom). (C) Heatmap showing top TF active in LP and HP population at day 7 (right). TF activity derived from scRNA data using Dorothea and Viper packages. (D) PCA plot showing expression of Myc (top), E2f (middle), and pigmentation genes (bottom) in the artificial cells simulated using RACIPE. (E) Violin plot showing Mitf expression in differentiating, mature, native, and proliferative melanocyte states in the progressive pigmentation model. (F) Violin plot showing Mitf expression in mature and proliferative states across different scRNA-seq data sets: human epidermal melanocytes (top), day 7 of the progressive pigmentation model (middle) and days 0, 3, and 5 of the progressive pigmentation model. All numerical data are listed in S1 Data.
(DOCX)

**S1 Table. List of differentially expressed cluster markers.**
(XLSX)

**S2 Table. ICC and FACS data quantitation table.**
(XLSX)

**S1 Data. All numerical data is listed that was used to generate plots shown in main and supplementary figures.**
(XLSX)

**S1 Raw Images. Full blot images for the blots represented in the study.**
(PDF)

## Acknowledgments

We thank Partha Chattopadhyay, Shweta Sahni, and Pooja Sharma for help with single-cell sequencing.

## Author Contributions

**Conceptualization:** Ayush Aggarwal, Rajesh S. Gokhale, Vivek T. Natarajan.

**Data curation:** Ayush Aggarwal.

**Formal analysis:** Ayush Aggarwal, Sarthak Sahoo.

**Funding acquisition:** Vivek T. Natarajan.

**Investigation:** Ayush Aggarwal, Ayesha Nasreen, Babita Sharma, Keerthic Aswin, Vivek T. Natarajan.

**Methodology:** Ayush Aggarwal, Sarthak Sahoo, Mohammed Faruq, Rajesh Pandey, Mohit K. Jolly, Abhyudai Singh, Vivek T. Natarajan.

**Resources:** Mohammed Faruq, Rajesh Pandey, Vivek T. Natarajan.

**Supervision:** Mohit K. Jolly, Abhyudai Singh, Vivek T. Natarajan.

**Validation:** Ayush Aggarwal.

**Visualization:** Ayush Aggarwal, Ayesha Nasreen, Sarthak Sahoo, Keerthic Aswin.

**Writing – original draft:** Ayush Aggarwal, Vivek T. Natarajan.

**Writing – review & editing:** Ayush Aggarwal, Ayesha Nasreen, Keerthic Aswin, Vivek T. Natarajan.

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
