## [Editor Report · Decision Letter 0]

20 Sep 2023

Dear Dr Natarajan, 

Thank you for submitting your manuscript entitled "Stochastic segregation of proliferation and maturation programs into distinct subpopulations enables rapid and sustainable pigmentation response" for consideration as a Research Article by PLOS Biology.

Your manuscript has now been evaluated by the PLOS Biology editorial staff as well as by an academic editor with relevant expertise and I am writing to let you know that we would like to send your submission out for external peer review.

Once your full submission is complete, your paper will undergo a series of checks in preparation for peer review. After your manuscript has passed the checks it will be sent out for review. To provide the metadata for your submission, please Login to Editorial Manager (https://www.editorialmanager.com/pbiology) within two working days, i.e. by Sep 22 2023 11:59PM.

Kind regards,

Ines

--

Ines Alvarez-Garcia, PhD

Senior Editor

PLOS Biology

---

## [Decision Letter · Decision Letter 1]

20 Dec 2023

Dear Dr Natarajan,

Thank you for your patience while your manuscript entitled "Stochastic segregation of proliferation and maturation programs into distinct subpopulations enables rapid and sustainable pigmentation response" was peer-reviewed at PLOS Biology. Please also accept my sincere apologies for the delay in providing you with our decision. The manuscript has now been evaluated by the PLOS Biology editors, an Academic Editor with relevant expertise, and by two independent reviewers. 

The reviews are attached below. As you will see, the reviewers find the conclusions interesting and significant for the field, however they also raise several concerns that would need to be addressed before we can consider the manuscript for publication. Reviewer 1 mentions that the manuscript should be clearer regarding novelty and general implications of the conclusions. Reviewer 2 is more critical and thinks that the cellular subpopulations have been only identified based on computational analysis of scRNA-seq data, and that independent confirmation at the protein level is needed to confirm the conclusions. This reviewer also thinks that the figures and figure legends should be improved so the readers can understand the content without having to look back at the text.

In light of the reviews, we would like to invite you to revise the work to thoroughly address the reviewers' reports. Given the extent of the revision needed, we cannot make a decision about publication until we have seen the revised manuscript and your response to the reviewers' comments. Your revised manuscript is likely to be sent for further evaluation by all or a subset of the reviewers.

**IMPORTANT - SUBMITTING YOUR REVISION**

3. Resubmission Checklist

a) *PLOS Data Policy*

b) *Published Peer Review*

d) *Blurb*

Please also provide a blurb which (if accepted) will be included in our weekly and monthly Electronic Table of Contents, sent out to readers of PLOS Biology, and may be used to promote your article in social media. The blurb should be about 30-40 words long and is subject to editorial changes. It should, without exaggeration, entice people to read your manuscript. It should not be redundant with the title and should not contain acronyms or abbreviations. For examples, view our author guidelines: https://journals.plos.org/plosbiology/s/revising-your-manuscript#loc-blurb

Sincerely,

Ines

--

Ines Alvarez-Garcia, PhD

Senior Editor

PLOS Biology

Reviewers' comments

Rev. 1:

The authors examined melanocytes derived from multiple sources, including human, zebrafish, and mouse cell lines, using scRNA seq multi-omics. The analyses identified the pigment (mature) and proliferation states, reflecting the heterogeneity of melanocyte cell states. The analyses look overall reliable, and I had only a few questions/comments for clarification (see below). However, the novelty and implications of the conclusion could have been clearer. Was it unexpected to see the heterogeneity of the cell state among melanocytes in humans and zebrafish or in culture? If it was an expected finding, what would this result add to the advancement of the field? Clarifying these will be helpful for the general readers.

The authors convey the melanocyte response to UV radiation in the introduction. However, it needs to be clear whether the cell states they analyze are related to UV response or homeostasis. What percentage of melanocytes proliferate and show pigment state under homeostasis and after sun exposure? The portion of each cell state is expected to be distinct between different physiological conditions.

Figure 1-5: Can the authors show representative pictures for high- or low-pigmentation melanocytes?

Figure 1E: The result description does not correspond to the figure.

Figure 5F, G should be labeled correctly.

Rev. 2:

This is a manuscript entitled by Aggarwal et al. entitled "Stochastic segregation of proliferation and maturation programs into distinct subpopulations enables rapid and sustainable pigmentation response". In this manuscript, the authors address the question of whether a defined population of melanocytes may occupy more than one differentiation state. They propose that such heterogeneity might, for example, account for the distinct responses of proliferation and differentiation exhibited by melanocytes upon exposure to UV irradiation. To attempt to answer this question, they analyze multiple sets of single-cell expression data to explore melanocyte cell heterogeneity across species, conditions, and transformation state to define distinct population states. With this information, they develop a gene regulatory network (GRN) model which links the pigmentation transcription factor genes MITF and LEF1 and which they use to generate computationally the existence of distinct, steady-state differentiation states that are observed experimentally in their scRNA-Seq analyses.

The datasets used and acquired for this manuscript are as follows:

(1) Reanalysis of scRNA data from normal human epidermal melanocytes (NHEM) obtained from a previous melanocyte study and published in 2021 (Belote et al., GSE151091);

(2) EGAS00001002927, whose use is described on page 5 and in Figures S2A-C (but otherwise not referenced);

(3) Melanocytes from zebrafish embryos at 24 hpf which were analyzed by scRNA-Seq;

(4) Forskolin daily- and alternate daily-pulsed NHEMs to drive proliferation or pigmentation respectively per Malcov-Brog et al., 2018; scRNA-Seq analysis;

(5) scRNA-Seq analysis of murine B16 melanoma cells separated into low pigmentation and high pigmentation phenotypes (Figure 2A-2D); also used for trajectory analysis (Figure 2E,F)

(6) Additional correlation of mouse hair follicle scRNA-Seq data from Infarinato et al. (2020) (Fig. S3E)

(7) scMulti-omics (RNA- and ATAC-seq) sequencing of cells in the progressive pigmentation model (B16) at days 0, 3, and 5, determining differentially accessible regions (DARs);

(8) H3K27Ac ChIP-Seq of B16 cells at day 5 of the pigmentation model, determining differentially and

The DAR and DHAcR data was used to enhance an additional analysis of (5) above to infer the transcription factors that are active in the low pigmentation (LP) and high pigmentation (HP) states.

Some general comments are as follows:

(1) The fundamental weakness of the manuscript in its current form is a lack of rigor and lack of independent validation (i.e. insufficient validation) of the characterizations of cellular subpopulations that are described after the experimental perturbations described above. This is especially true for (4) (same- and alternate-day fsk-pulsed NHEMs) and (5) (scRNA-Seq analysis of B16 melanoma cells). The authors have not independently validated the existence of distinct cell subpopulations at the cellular level by examining individual cells for their expression or non-expression of the specific markers which can distinguish the subpopulations they claim are revealed by the scRNA data and subsequent analytical dimensionality reduction. Although this validation could be performed in different ways, one method would involve using markers that might distinguish the subpopulations (i.e. MITF or TYR, MKI67, and TWIST1) and immunostaining cells to score them for non-overlapping, high level expression of these markers. For example, to validate the data, one would expect that the proportions of MITF+/MKI67-/TWIST1-, MITF-/MKI67+/TWIST1-, and MITF-/MKI67-/TWIST1+ cells in a population, scored for expression after immunofluorescence, should correspond to the proportions of those populations described in the scRNA-Seq analysis. The important conclusions of this manuscript are predicated upon the existence of these cellular subpopulations; however, in the current version of the manuscript, they are only demonstrated on the basis of computational analysis of scRNA-Seq data. Since the interpretation of scRNA-Seq is subject to the application of discretionary and what can be arbitrary parameters and constraints applied to the high-dimensional data, in my opinion independent confirmation at the protein expression level of these subpopulations is necessary to accept the conclusions of the manuscript as valid.

(2) In my opinion, the figures and figure legends are incompletely annotated. It should be possible at a minimum to appreciate the overall content of a figure and its legend simply from viewing the figure and the legend together, without constant reference to the manuscript text. However, that is not possible with this manuscript. There are too many specific examples to describe completely, but one example is Figure 4 and its legend. It is not apparent from reviewing the figure and legend together that the data represents the results of scATAC-seq and ChIP-Seq experiments. These facts are not directly stated in the figure or the legend. I am not going to offer specific suggestions about the enhanced level of detail that needs to be provided to understand the figures and legends adequately, since this should represent common scientific writing practices, but in general the dearth of information applies to all figures and legends and they all must be improved to render the manuscript more comprehensible.

Some specific comments follow:

1. p.3, introduction - There seems to be some unfounded assumptions in the introductory statements provided about the stress and proliferation responses. The authors state that the "stress response (which) includes proliferation, … , to prevent depletion of the melanocyte population". To impute that proliferation occurs specifically to prevent depletion of the melanocyte population appears to me to be an unsupported statement, with no direct evidence to prove that the "reason" for the stress-induced proliferation is to prevent depletion versus, for example, simply to provide a larger number of melanocytes to enhance photoprotection. Similarly, the authors imply that "the pigmentation process", through its induction of a "terminal differentiation state" in turn "reduces the likelihood of cell proliferation upon subsequent UV exposure". Again, I know of no direct evidence to support these statements. Senescence might reasonably reduce the propensity for subsequent cell proliferation, but I am not aware of any absolute inverse relationship between "differentiation" and "likelihood of cell proliferation". The authors should strive to limit their statements and conclusions to those that can be supported by evidence.

2. Also p.3 - the authors refer to studies which "suggest the existence of melanocyte states other than the terminally differentiated pigmented state in the epidermis", following with "the coexistence of such transcriptionally distinct melanocyte sub-populations in the same environment is not yet known" and that their "interchange between states remains speculative". The authors seem not to have considered work described in Sun Q et al. (2023) Nature 616, 774 and Joshi et al. (2019) PLoS Genet 15(4):e1008034 that describes the existence of melanocytic cells in the same or similar environments in distinct differentiation states and their interconversion between states.

3. Also p. 3 - unsupported, unreferenced statements such as "Heterogeneity within a seemingly homogeneous population is considered to provide the means to maintain the diversity required for the survival of the overall population" should be either avoided or fully substantiated by references.

4. P. 4 - by "generic feature' do you mean "inherent feature"?

5. Supplementary Figure 1 - the authors refer to the "stacked bar plot" for panel B, but to me panel B looks like a UMAP plot, not a stacked bar plot. The authors should make this correction and determine whether any similar corrections in Figure legends are required.

6. P. 5, bottom - the authors refer to "another publicly available human dataset", but the source and nature of the dataset is not specified, neither here nor in the Methods section.

7. P. 6 and Figure S2A - assuming that the other "publicly available human dataset " exists, as questioned above, it is strange that the authors simply list the top differentially expressed genes between the clusters in Figure S2A, and make no attempt to integrate this data with the data presented in Figure S1 (Belote et al.) to determine whether the different clusters identified in each separate dataset instead have shared features between datasets. In other words, does Cluster 0 in Figure S1 correlate highly with Cluster 1 (or any other numbered cluster) in Figure S2? It seems as if trying to answer this question, with the data and tools available to you, should be a greater priority than simply performing another assessment of melanocytic heterogeneity in zebrafish as the authors have done.

8. p.5, Figures S2A-C; the authors describe the use of EGAS00001002927, but I cannot find a reference to the manuscript or other resource that contains this dataset. The authors need to provide this information.

9. pp.6-7, experiment with forskolin (Fsk)-pulsed NHEMs: Which clusters correspond to which set of treated cells? (When initially described as clusters corresponding to the forskolin daily vs. alternate-day pulsed cells, which number clusters correspond to which treatment? Why isn't this stated up front in the text?

10. P.6 (fsk-pulsed NHEM experiment) - of greater concern is the assumption made about fsk-pulsed NHEM, that their behavior will mimic the behavior of in vivo melanocytes and in vitro (cultured) MNT-1 cells from Malcov-Brog et al. (2018). Malcov-Brog et al. (2018) did not perform this experiment with NHEMs. The authors have not first independently validated the findings of Malcov-Brog et al. with NHEMs, namely that an alternate-day pulse with either UV or fsk results in greater pigmentation and expression of pigmentation genes, and a dampened oscillation of Mitf expression levels, as was shown with MNT-1 melanoma cells in Malcov-Brog et al. Hence the authors need to confirm similar basic findings with NHEM before they can build upon these conclusions with the single-cell expression analysis and clustering that they do in this manuscript.

11. p. 11 - the "progressive pigmentation model" is not well-defined.

12. p. 12 - why can't NHEMs also be used to trace dynamics of state transitions?

13. p. 13 - did Raja et al. (2020a) actually show that H3K27Ac selectively activated pigmentation genes?

14. p.15 - Principal component analysis, not "principle"

15. p.15 - "all the three independent analyses described above" - please be specific and name the analyses.

16. In the colony analysis (p.10, Figure 3), how can the authors be sure they are capturing all of the cells optically when there is no specific cellular marker used?

Data available under GEO as GSE233198.

---

## [Decision Letter · Decision Letter 2]

24 Jun 2024

Dear Dr Natarajan,

Thank you for your patience while we considered your revised manuscript entitled "Stochastic segregation of proliferation and maturation programs into distinct subpopulations enables rapid and sustainable pigmentation response" for publication as a Research Article at PLOS Biology. This revised version of your manuscript has been evaluated by the PLOS Biology editors, the Academic Editor and one of the original reviewers.

Based on the review (attached below), we are likely to accept this manuscript for publication, provided you satisfactorily address the remaining points raised by Reviewer 2. Please also make sure to address the data and other policy-related requests stated below.

In addition, we would like you to consider a suggestion to improve the title:

"Distinct melanocyte subpopulations defined by stochastic expression of either proliferation or maturation programs enable a rapid and sustainable pigmentation response to UV radiation"

We expect to receive your revised manuscript within two weeks. 

*Published Peer Review History*

*Press*

Sincerely,

Ines

--

Ines Alvarez-Garcia, PhD

Senior Editor

PLOS Biology

DATA POLICY:

Many thanks for submitting the files containing all individual quantitative observations that underlie the data summarized in the figures. However, we are still missing data from the following figures - please provide these:

Fig. 1B, D, E, F, G, I; Fig. 2A, C; Fig. 3A, D, E; Fig. 4B, D; Fig. 5B, D-G; Fig. S1A-E; Fig. S2A, D, E; Fig. S5C-E; Fig. S7A-D, F, G and Fig. S8A-F

Please also ensure that figure legends in your manuscript include information on WHERE THE UNDERLYING DATA CAN BE FOUND.

Please also ensure that your Data Statement in the submission system accurately describes where your data can be found.

**Please make sure the data you have deposited in GEO under the accession GSE233198 is made publicly available at this stage.

CODE POLICY

Reviewers' comments

Rev. 2:

This is the second revision of a manuscript entitled "Stochastic segregation of proliferation and maturation programs into distinct subpopulations enables rapid and sustainable pigmentation response" by Aggarwal et al. This manuscript addresses the rapid and distinct responses of proliferation and differentiation that melanocyte exhibit upon exposure to UV irradiation by demonstrating the existence of multiple, discrete differentiation states in murine and human melanocytes and melanoma cells. As noted before, this is accomplished both by a reanalysis of previous in vivo single-cell expression data from Belote et al.; and through acquisition and analysis of single-cell expression data from zebrafish melanocytes, forskolin-pulsed normal human epidermal melanocytes (NHEM), murine B16 melanoma cells, human MNT-1 melanoma cells, and other datasets.

The investigators have addressed the major criticism of this reviewer, which was inadequate validation of the distinct transcriptional states inferred from their initial gene expression results at the protein level. To their credit, the investigators attempted to validate the effects of daily versus alternate-day forskolin on normal human epidermal melanocytes (NHEM) compared to effects on MNT-1 melanoma cells that had been previously reported. The results of the validation experiment show a modest skewing of a flow cytometry curve of alternate-day melanocytes toward higher melanin content, a result consistent with the notion that alternate-day forskolin promotes development of a more highly pigmented melanocyte. For example, they have now nicely used immunofluorescence and quantitative cellular fluorescence to demonstrate an inverse relationship between the expression of the differentiation marker Tyr and the proliferation marker Ki67 in NHEM (Figure S3), providing further evidence of the characteristics of the HP and LP populations developed through differing administration schemes of forskolin to these cells. Similar cellular and protein level validation for MNT-1 cells in presented in Figure S4. Figure S6 provides cellular level and protein validation, using proliferation and differentiation markers, of B16 melanoma cells utilized in the progressive pigmentation model.

Some additional points follow:

1. Introduction - "when the pigmentation process leads to a terminal

differentiation state, which could reduces the likelihood of cell proliferation due to the

burden on pigmentation(Bennett, 1983)". This construction remains awkward.

2. Introduction - "Manipulating the α-MSH pathway using a pharmacological activator," - unclear why there is the need for a comma after "activator"

3. At some point in the manuscript, the authors should explain how the side scatter population corresponds to cellular pigmentation (unless the explanation is present and I have overlooked it)

4. Results, newly added section on B16 cell immunofluorescence: "existence" rather than "existance".

5. Discussion: may want to refer to the "B16 progressive pigmentation model", just to keep the distinction between this model and the other experimental models used in the research (NHEM, MNT-1) fully distinct for the reader.

6. Methods: should be "Alexa Fluor", not "Alexa Flour". Also, I think it is "principal components", not "principle components".

7. Methods: Did the authors omit the description of techniques for the cellular and protein validation analysis of the MNT-1 cells?

8. References: do not see the Natarajan et al. 2024 reference listed in the Bibliography.

---

## [Editor Report · Decision Letter 3]

30 Jul 2024

Dear Dr Natarajan,

Thank you for the submission of your revised Research Article entitled "Distinct melanocyte subpopulations defined by stochastic expression of proliferation or maturation programs enable a rapid and sustainable Pigmentation response" for publication in PLOS Biology. On behalf of my colleagues and the Academic Editor, Colin Jamora, I am delighted to let you know that we can in principle accept your manuscript for publication, provided you address any remaining formatting and reporting issues. These will be detailed in an email you should receive within 2-3 business days from our colleagues in the journal operations team; no action is required from you until then. Please note that we will not be able to formally accept your manuscript and schedule it for publication until you have completed any requested changes.

PRESS

Sincerely, 

Ines

--

Ines Alvarez-Garcia, PhD

Senior Editor

PLOS Biology
